# Effect of Zwitterionic Additive on Electrode Protection through Electrochemical Performances of Anatase TiO_2_ Nanotube Array Electrode in Ionic Liquid Electrolyte

**DOI:** 10.3390/ijms24043495

**Published:** 2023-02-09

**Authors:** Aleksandra Roganović, Milan Vraneš, Nikola Cvjetićanin, Xiaoping Chen, Snežana Papović

**Affiliations:** 1Faculty of Sciences, Department of Chemistry, Biochemistry and Environmental Protection, University of Novi Sad, Trg Dositeja Obradovića 3, 21000 Novi Sad, Serbia; 2Faculty of Physical Chemistry, University of Belgrade, Studentski trg 12-16, 11158 Belgrade, Serbia; 3Hangzhou Bay Automotive College/Mechanical College, Ningbo University of Technology, No.769, Bihai Road 2, 315336, Hangzhou Bay New Zone, Ningbo 315000, China

**Keywords:** imidazolium ionic liquid, functionalized additive, TiO_2_ nanotubes, spectroscopy, galvanostatic cycling

## Abstract

In this work, a functionalized zwitterionic (ZI) compound 1-butylsulfonate-3-methylimidazole (C_1_C_4_imSO_3_) was synthesized and tested as an additive to LiTFSI/C_2_C_2_imTFSI ionic liquid-based electrolytes for lithium-ion batteries. The structure and purity of C_1_C_4_imSO_3_ were confirmed by NMR and FTIR spectroscopy. The thermal stability of the pure C_1_C_4_imSO_3_ was examined by simultaneous thermogravimetric–mass spectrometric (TG–MS) measurements and differential scanning calorimetry (DSC). The LiTFSI/C_2_C_2_imTFSI/C_1_C_4_imSO_3_ system was tested as a potential electrolyte for lithium-ion batteries by using anatase TiO_2_ nanotube array electrode as the anode material. This electrolyte with 3% C_1_C_4_imSO_3_ showed significant improvement of lithium-ion intercalation/deintercalation properties, such as capacity retention and Coulombic efficiency compared to electrolyte without additive.

## 1. Introduction

Lithium-ion batteries (LIBs) are becoming increasingly essential for electrifying the transportation systems for sustainable mobility in the future. However, large-scale application of lithium-ion batteries is limited by serious safety concerns when the LIBs are exposed to thermal, mechanical, or electrical abuse conditions [1,2]. Their high energy density associated with the use of volatile, flammable electrolytes based on organic solvents creates safety risks [3,4]. Since the electrolyte is the component that connects all parts, it notably impacts a lot of technological and chemical aspects of LIBs.

To overcome these challenges, modifications of the electrolyte through substituting the organic solvent or addition of functional additives to the electrolyte, such as ionic liquids (ILs), have been discussed [5,6,7,8,9,10,11,12,13,14,15]. From a literature survey about the application of ILs in LIBs, it was noticed that the most efficient anion was *bis*(trifluoromethylsulfonyl)imide ([NTf_2_]^−^) due to its delocalized charge [8,9]. Potential optimal electrolytes for lithium-ion batteries being studied are organic solvent/ionic liquid mixtures that combine high thermal stability, low flammability, and high electrical conductivity [11,12].

To overcome the safety concerns related to interactions between conventionally used electrolytes and electrode materials, researchers have developed functionalized electrolyte systems in combination with robust electrodes. Electrode materials that enhance the battery’s capacity and energy density could be TiO_2_ nanotube arrays (NTAs) [16,17,18,19,20,21,22,23,24,25]. TiO_2_ nanotubes are specific materials, which can combine with current collector and electrode material. TiO_2_ is a robust and resistant material, allowing on the use of higher temperatures because there is no possibility of removing materials from the electrodes at higher temperatures [16,17,18,19,20]. In addition, such NTA electrodes enable immediate, direct recording of Raman and Fourier Transform Infrared (FTIR) spectra on their surface.

Current demands for LIB electrolytes include developing electrolytes containing components that synchronously achieve different functions, such as electrolyte stabilization towards the electrode materials and electrode functionalization [26,27,28,29,30]. As a result, the knowledge based on ionic liquids induced the synthesis of novel families of compounds established on them—the zwitterionic compounds, because it is assumed that there is a possibility of a stabilization effect towards the electrode [31,32,33,34,35,36,37,38]. One of the imperatives of our research is thermal and safety improvement of LIBs. This research study examined the electrochemical properties of 0.5 mol·dm^−3^ LiTFSI (Figure 1a) in ionic liquid 1,3-diethylimidazolium *bis*(trifluoromethylsulfonyl)imide, C_2_C_2_imTFSI (Figure 1b) with 3% a functionalized additive (zwitterionic compound 1-butylsulfonate-3-methylimidazole C_1_C_4_imSO_3_; ZI, Figure 1c), along with the performance of anatase Ti/TiO_2_ nanotube array (NTA) cells. For this purpose, the zwitterionic compound C_1_C_4_imSO_3_ was synthetized in his work, as shown in Figure 1d.

The electrochemical testing was conducted to compare the stability of the electrolyte studied in this work with that in a previous study [28], which did not contained the zwitterionic compound C_1_C_4_imSO_3_ as an additive. In comparison with their IL counterparts, the present ZI had a more polar character, suggesting that this compound may become attractive in battery electrolytes. A comprehensive consideration of the ZI compound’s electrochemical behavior and mechanism is needed for deeper thermochemical and physicochemical profiling.

## 2. Results and Discussion

### 2.1. Thermogravimetric and DSC Analysis

The thermal properties of pure synthesized zwitterionic salt C_1_C_4_imSO_3_ were determined by thermogravimetric (TG) and differential scanning calorimetry (DSC) measurement. The obtained TG and DSC decomposition curves of C_1_C_4_imSO_3_ are shown in Figure 2.

From the presented thermogravimetric curve (Figure 2, black line), the synthesized C_1_C_4_imSO_3_ was stable up to 330 °C, implying its high thermal stability. The TGA thermograms in Figure 2 did not show any significant weight loss near 100 °C, demonstrating that this zwitterion is not hygroscopic. From TG–DSC measurements, we observed the softening of the C_1_C_4_imSO_3_ compound with loss of weight and accompanied with an endothermic peak in the simultaneous DSC curve (Figure 2, blue line), demonstrating the melting and decomposition process [20,21]. The DSC curve (Figure 2, blue line) showed one sharp peak around *T*_melt_ = 236 °C. The second DSC peak corresponds to the complete decomposition of C_1_C_4_imSO_3_ (around *T* = 365 °C). The comparison of the *T*_melt_ to that of similar zwitterionic compounds, regarding the length of the spacer between the cation and anion group, was in accordance with other research [31]. Comparison of the sulfonate anion and imidazole cation part with other compounds presented in the work of Yoshizava et al. [31] and Galin et al. [36], where the only difference in length is in the molecule part that connects them, suggests that longer spacer indicate lower melting points. This is probably because of the flexibility increase with increasing spacer length. Figure 3 illustrates the DSC thermogram of the synthesized C_1_C_4_imSO_3_ compound in an extended lower temperature range.

The obtained solid compound exhibited a prominent melting point upon heating (*T*_m_ = 236 °C) as the top of an endothermic peak. In the literature, zwitterions which have sulfonate with a longer spacer between the cation and anion show lower *T*_m_ values at the range of spacer length studied [31]. The DSC data showed that the zwitterionic compound based on imidazolium cations has a significantly higher melting point than those of common imidazolium-based ILs. In this zwitterionic structure, the cation and anion are tethered, and therefore contribute to restricting the molecule’s vibrational and rotational motions [31]. Therefore, the negative charge is delocalized by the sulfonate group electron-withdrawing effect, and the interaction force between the anion and cation is lower than in the case of sulfonate anions [31,34].

### 2.2. Thermogravimetry Coupled with Mass Spectrometric (TG–MS) Analysis

We performed additional thermal characterizations of the synthetized C_1_C_4_imSO_3_ using thermogravimetry coupled with the mass spectrometry (TG–MS) because it does not exist in the research literature. During the entire process of degradation, the evolution of the essential fragment products were collected by the TG–MS technique. From Figure 4 we can see the thermochemical decomposition in one weight loss step, the first derivative curve with time (DTG), as well as the TG–MS fragment ion intensities for formed products. The temperature interval from 320 to 390 °C mostly shows the mass loss of C_1_C_4_imSO_3_.

The high intensity of the MS peaks for ions with *m*/*z* = 18 (H_2_O) and *m*/*z* = 17 (OH) are shown as sharp DTG peaks at 380 °C. The intensity ratio of fragments with *m*/*z* = 18 (H_2_O): *m*/*z* = 17 (OH) = 5:1, is in accordance with the National Institute of Standards and Technology (NIST) mass spectral data [39,40,41,42,43], that is attributed to these fragments. As Figure 4a implies, a relatively large amount of H_2_O (*m*/*z* = 18) and OH (*m*/*z* = 17) was formed during the recombination of the atoms and heating, due to the oxygen atoms from the structure of C_1_C_4_imSO_3_.

Characteristic for this thermal decomposition is the departure of products with fragments *m*/*z* = 28 (CO) and 32 (O_2_). They are frequently formed from the removal of the carbon and its recombination with oxygen leaving CO *m*/*z* = 28. That process occurs at 375 °C, as seen in Figure 4b. The removal of SO and SO_2_ from C_1_C_4_imSO_3_ released during heating (*T* = 370 °C) is indicated with the (*m*/*z* = 48), and SO_2_ (*m*/*z* = 48) is released at *T* = 375 °C. The *m*/*z* = 48 and *m*/*z* = 64 indicate that two overlapping steps occurred in the range *T* = 370–375 °C and were difficult to distinguish (Figure 4c).

The products with fragments *m*/*z* = 14 and 16 are ingrained in this peak at 370 °C: *m*/*z* = 14 (N) and *m*/*z* = 16 had a similar shape (NH_2_) with an equally low magnitude (Figure 4d). The *m*/*z* = 14 and *m*/*z* = 16 fragments started around 375 °C. The C-C and C-H bonds broke to form free radicals, and recombined into small fragments with the increase in temperature. Accordingly, the most significant fraction of *m*/*z* = 36 can be attributed to C-C-C (C_3_) from a side chain attached to the imidazole ring of C_1_C_4_imSO_3_. With the increase in temperature, the C-C and C-H bonds broke to form free radicals, which are recombined into more considerable fragments as *m*/*z* = 44 (CH_3_-CH_2_-CH_3_), that were detected with lower intensities (Figure 4e).

With further heating (*T* = 407 °C), one of the highest intensities for the sample was released and correlates to the removal of H from C_1_C_4_imSO_3_ (*m*/*z* = 1; Appendix A). Appendix A summarizes the identification results and normalized ion currents for the thermal decomposition of the pure compound in this work.

### 2.3. Flammability of Electrolyte

Regarding the electrolyte safety aspect, it is essential to estimate the flammability of 0.5 mol·dm^−3^ LiTFSI/C_2_C_2_imTFSI/C_1_C_4_imSO_3_ electrolyte. A flame did not appear during 120 s of heating, which indicates that the LiTFSI/C_2_C_2_imTFSI/C_1_C_4_imSO_3_ electrolyte is nonflammable (Appendix A). The flammability of the electrolyte with same concentration of lithium salt in ionic liquid electrolyte was investigated in a previous publication [28].

### 2.4. Electrochemical Characterization

#### 2.4.1. Galvanostatic Cycling

The discharge/charge capacities during Li^+^-ion insertion/deinsertion into/from TiO_2_ NTAs for electrolyte LiTFSI/C_2_C_2_imTFSI and for the same electrolyte with the zwitterion additive LiTFSI/C_2_C_2_imTFSI/C_1_C_4_imSO_3_, were obtained at a current density of 50 µA/cm^2^ and are shown in Figure 5a–c, respectively. For the LiTFSI/C_2_C_2_imTFSI electrolyte without additive, the initial discharge/charge capacity of the Ti/TiO_2_ NTA electrode was 370/303 mAh·g^−1^. After seven cycles and increase in the charge capacity, both the discharge and charge capacities decreased duslightring further cycling, (Figure 5a). Although the capacity decreased with the increase in the number of GS cycles, the stabilization of the capacity was not achieved after 100 cycles, when the discharge/charge capacity was 228.0/217.3 mAh·g^−1^. The Coulombic efficiency increase during cycling and amounted 93.7% after 50 cycles, 94.6% after 75 cycles, and 95.3% after 100 cycles. The electrolyte at the beginning was colorless, but after 100 cycles it became a pale yellow, which shows the presence of electrolyte decomposition (Figure 5c). The increase of Coulombic efficiency as the number of cycles increased, indicates that the electrolyte decomposition process was more pronounced at the beginning of cycling, but continued during further GS cycling.

In the case of the electrolyte with additive, LiTFSI/C_2_C_2_imTFSI/C_1_C_4_imSO_3_, the initial discharge capacity of the Ti/TiO_2_ NTAs electrode was 325.4 mAh·g^−1^, and the charge capacity was significantly lower, at 231.8 mAh·g^−1^ (Figure 5b). During the initial 14 cycles, the charge capacity increased slightly. Then, both discharge and charge capacities decreased until the 43^rd^ cycle, when the decrease of both capacities stopped and their values became almost identical. Further GS cycling even led to a slight increase in capacity. The Coulombic efficiency was 99.0% after 43 cycles, 99.2% after 50 cycles, 99.5% after 75 cycles, and 99.6%after 100 cycles. After 100 cycles the final discharge/charge capacity was 249.4/250.5 mAh·g^−1^ and the electrolyte remained colorless. The presence of the zwitterion additive C_1_C_4_imSO_3_ led to the formation of a film on the working electrode, with this process being completed in ~40 cycles. This prevented further decomposition of the electrolyte and allowed the Li^+^-ion insertion/deinsertion process to occur with high Coulombic efficiency, and without capacity fade during further cycling. The comparison of the capacity of the Ti/TiO_2_ NTAs electrode in the electrolyte with and without additive can be seen in Figure 5c.

The voltage profiles, for the 100th GS cycle, during lithiation/delithiation of the Ti/TiO_2_ NTA electrode in the LiTFSI/C_2_C_2_imTFSI are presented in Figure 5d. The region in discharge during lithiation, where the voltage profile was relatively flat (~1.74 V in the middle part), designates the existence of two phases: the lithium poor Li_0.026_TiO_2_ and lithium rich Li_0.52_TiO_2_ phases [44,45,46]. When the transition from the first to the second phase during lithiation was completed, the inclined part of voltage profile started. This part ended at the lower cut-off voltage of 1.0 V, and can be attributed to surface storage mechanisms [45,46]. It can be seen in Figure 5d that in the case of the LiTFSI/C_2_C_2_imTFSI electrolyte, the contribution of the inclined part of the voltage profile to the overall capacity was larger from flat part, while for the electrolyte with additive, LiTFSI/C_2_C_2_imTFSI/C_1_C_4_imSO_3_, it is vice versa. This shows that the film formed during GS cycling at the surface of the TiO_2_ NTs, due to the presence of the zwitterion additive C_1_C_4_imSO_3_, decreased the surface storage, but has the effect of increasing the depth of lithium insertion into the TiO_2_ NTs. This increased of the overall capacity of the active electrode in the electrolyte with additive comparing to the electrolyte without additive, both in discharge and charge as shown in Figure 5d for the 100th cycle. The increase in overall capacity, together with the stabilization of the discharge/charge capacity with high Coulombic efficiency and without capacity fade that is shown in Figure 5b,c, ultimately led to the excellent performance of the LiTFSI/C_2_C_2_imTFSI/C_1_C_4_imSO_3_ electrolyte.

After GS cycling, the UV-Vis spectra of the electrolyte solutions were recorded for both tested electrolytes, especially considering the observed change from colorless to yellow in the case of the LiTFSI/C_2_C_2_imTFSI electrolyte. The recorded UV-Vis spectra showed strong absorption by the newly formed compound(s), occurring at very low concentrations but were undetectable with UV-Vis spectroscopy (Appendix A).

The SEM micrographs showed no significant change in the morphology of TiO_2_ before and after galvanostatic cycling in the LiTFSI/C_2_C_2_imTFSI/C_1_C_4_imSO_3_ electrolyte, as shown in Figure 6.

### 2.5. Spectroscopy Measurements after Cycling

#### Vibrational Spectroscopy

The Ti/TiO_2_ NTA electrodes from both tested electrolytes after GS cycling were further analyzed with FTIR spectroscopy. First, the electrodes were washed with acetone to fully dissolve the ionic liquid C_2_C_2_imTFSI. The only sharp band for both electrolytes was located at ~1420 cm^−1^ (Figure 7a) and could be assigned to S=O stretching of the sulfate group (1415–1380 cm^−1^) rather than to the same vibration of sulfonates (1372–1335 cm^−1^) [47,48,49]. For the SO_3_ molecule the antisymmetric stretch was at 1391 cm^−1^ [50], which is also lower than the obtained value. The formation of surface SO_4_^2−^ may occur if surface SO_3_^2−^ reacts with lattice oxygen [51] when adjacent Ti^4+^ is reduced to Ti^3+^ during intercalation of Li^+^-ions. For the electrolyte with ZI additive, which has sulfonate group, such formation of surface SO_4_^2−^ seems to occur in a simpler way compared to the electrolyte without additive, where the SO_2_^2−^ group inside the TFSI^−^ anion is in a more complex coordination. Most likely, this is the reason why the intensity of the IR band at ~1420 cm^−1^ was higher in the presence of the ZI additive (Figure 7a). In the IR spectra of both electrolytes, on the high frequency side of the obtained IR band a shoulder exists at ~1480 cm^−1^ due to the presence of a wide band of lower intensity. Considering that the low intensity of this band composed of more overlapping bands and the existence of a curved base line which all make deconvolution problematic, the assignation of this band was not performed. However, it is important to stress that its intensity follows the intensity of the band at ~1420 cm^−1^.

The IR spectra showed that during GS cycling, decomposition of the ZI additive C_1_C_4_imSO_3_ and electrolyte LiTFSI/C_2_C_2_imTFSI occurred at the same surface sites of the TiO_2_ NTs. In the electrolyte where additive was present, its decomposition blocked surface sites for LiTFSI/C_2_C_2_imTFSI decomposition. This stabilized the electrode capacity after a certain number of cycles and enabled a high Coulombic efficiency of the Li^+^-ion insertion/extraction process. Without additive, the decomposition of the LiTFSI/C_2_C_2_imTFSI electrolyte continued during all 100 cycles without final stabilization of the electrode capacity which was dropping. The Coulombic efficiency gradually increased, but did not reach the values as in the electrolyte with additive. The IR spectra of the electrodes after an additional wash with water, which removed all molecules from the surface of the TiO_2_ NTs, are shown in Figure 7b.

The Raman spectra of the Ti/TiO_2_ NTA electrodes from both electrolytes were also recorded. The Raman modes at 635 cm^−1^ (E_g_), 519 cm^−1^ (A_1g_, B_1g_), 399 cm^−1^ (E_g_), 205 cm^−1^ (E_g_), and 152 cm^−1^ confirmed the presence of the TiO_2_ anatase phase for both electrodes. The low intensity Raman band at 325 cm^−1^, observed in the spectrum of the electrolyte with ZI additive was of slightly better quality and was a combination band [52,53] (Figure 8).

The results imply that the sulfonate-based zwitterionic compound might facilitate Li^+^ movement because a partial negative charge that developed at the end of the sulfonate SO_3_^−^ group is able to bind with Li^+^, leading to pronounced kinetic behaviors during the process of de-lithiation [33,34,35]. This could be assigned to the uniformity of the light artificial layer based on sulfonate at the surface of the TiO_2_ NTs [49], which could be liable for retarding the decomposition of the electrolyte. In some reported papers, previously investigated zwitterionic compounds, such as *N*,*N*-dimethylpyrrolidinium methyl sulfonate showed a minimization of electrolyte decomposition resulting in interfacial stability of the used electrodes [38]. In addition, some of zwitterionic compounds [1-(1-Butylpyrrolidinium)butane-4-sulfonate betaine, 1-(Tri-n-butylphosphonium)butane-4-sulfonate betaine [54]], contributed to the stabilization of the Li^+^ ions in the mixtures. When compounds with a similar sulfonate functional group, such as sodium dodecylbenzene sulfonate [55] interact with nanotubes, the self-organization of organic molecules with the carbon nanotubes led to the formation of functionalized molecular structures with advanced properties [55,56]. Some future steps for further development of efficient zwitterionic compounds could lie in the modification of substituents in the imidazole ring (such as a vinyl group, or oxygenated alkyl side chain attached at one N in the imidazole ring) with the same SO_3_ part from the other side of imidazole ring. The interfaces’ structure and the electrode-protecting films’ nature are other aspects that need appropriate consideration.

## 3. Materials and Methods

### 3.1. Materials

The preparation of the TiO_2_ nanotube arrays (NTAs) proceeded by anodic oxidation of Ti foil that is 0.25 mm thick and 0.5 cm wide (Table 1). Under the constant voltage of 30 V, in 0.7% NH_4_F in glycerol solution by using graphite as a cathode, the anodization was conducted. For a period of 6 h the anodic oxidation was accomplished in order to obtain a thick NT layer [57] for accurate mass determination. The complete electrode preparation was carried out in the manner described in our previous paper [28]. It should be noted that after the galvanostatic cycling experiments (described in Section 2.4.1) the SO_3_ group from the C_1_C_4_imSO_3_ zwitterionic compound from the electrolyte (detailed electrolyte content is described in next Section 3.1.1) was partially adsorbed at the TiO_2_ nanotubes used as the electrode in these experiments.

#### 3.1.1. Synthetic Procedure for the Zwitterionic Compound

The detail specifications of all reagents used in the experimental work are tabulated in Table 1. An equimolar amount of 1,4-butane sultone and 1-methylimidazole were mixed in acetonitrile (Figure 1d), heated, and stirred at temperature around 60 °C for 12 h. Afterwards, the used solvent was removed in vacuo and remaining solid was washed with 8 × 20 mL acetonitrile. It was dried in vacuo to give 1-butylsulfonate-3-methylimidazole (yield, 74%) at room temperature as a white solid. The resulting white solid, C_1_C_4_imSO_3_, was heated under a vacuum and stored in the vacuum desiccator with P_2_O_5_, with a final yield of 90%. The obtained zwitterionic compound was stored in the dry box under an atmosphere of nitrogen (the structure is presented in Figure 1c).

The structural confirmation of the synthesized zwitterion C_1_C_4_imSO_3_ was carried out by ^1^H, ^13^C NMR, and FTIR analyses (Appendix A show the recorded spectra together with assignments in Appendix A). The purity of the C_1_C_4_imSO_3_ was estimated from the ^1^H NMR spectra (Table 1).

Before use, the ionic liquid C_2_C_2_imTFSI was stored in a vacuum desiccator and in the dry box under an atmosphere of nitrogen. Using the Metrohm 831 Karl Fischer coulometer, the water content in IL was determined (Table 1).

Lithium salt, LiTFSI, was dried in a vacuum at *T* = 110 °C. Under inert conditions in an argon-filled glove box, the electrolyte was used to dissolve the LiTFSI in ionic liquid C_2_C_2_imTFSI, obtaining a Li^+^ concentration *c*(Li^+^) of 0.5 mol·dm^−3^. For simplicity, the electrolyte 0.5 mol·dm^−3^ LiTFSI in C_2_C_2_imTFSI was referred as LiTFSI/C_2_C_2_imTFSI in this work.

The other investigated electrolyte was obtained by dissolving 0.5 mol·dm^−3^ LiTFSI in a previously made mixture of ionic liquid C_2_C_2_imTFSI and ZI compound C_1_C_4_imSO_3_, where 3% of C_1_C_4_imSO_3_ was added.

### 3.2. Apparatus and Procedures

#### 3.2.1. Spectroscopy Measurements (NMR, FTIR, and UV-Vis)

NMR. Spectroscopic identification of the synthesized C_1_C_4_imSO_3_ using nuclear magnetic resonance (NMR) was recorded in D_2_O solvent at temperature *T* = 25 °C on a Bruker Advance III 400 MHz spectrometer. For assignation of 13C NMR spectra, the selective decoupling method was used and for 1H homodecoupling the 2D COSY method was used.

FTIR. A Shimadzu Fourier Transform Infrared (FTIR) Spectrometer Reflectance with a Universal Diamond ATR Sampling Accessory (MIRacle 10 ATR; Dia/ZnSe) was used to perform the FTIR measurements at room temperature in a range from 500 to 3400 cm^−1^. The synthesized C_1_C_4_imSO_3_ or the tested samples during the experimental work were placed on the top of the diamond crystal and FTIR was performed under an inert atmosphere.

UV-Vis. In the wavelength range of 250–800 nm, the absorption spectra of the investigated electrolytes in this work were recorded. The UV-Vis measurements were conducted on a Thermo Scientific UV-Vis spectrophotometer Evolution 220 in 1 cm path length quartz cuvettes, at room temperature and under an inert atmosphere.

Raman spectroscopy. Raman spectra of the TiO_2_ nanotubes were collected using a solid-state Nd:YAG laser excitation line of 532 nm, with an incident laser power of less than 60 mW to minimize the heating effects on the samples. The measurements were performed at room temperature. A Tri Vista 557 triple spectrometer coupled with a nitrogen-cooled CCD detector was used. The measurement was recorded in the wavenumber range of 1400–50 cm^−1^.

#### 3.2.2. Thermal Properties (TG, TG-MS and DSC)

TG measurement. The thermophysical properties of the synthesized C_1_C_4_imSO_3_ were analyzed using a simultaneous thermogravimetric analyzer with a differential scanning calorimetry TG/DSC thermal analyzer SDT Q600 (TA Instruments, Milford, Massachusetts, USA) with calorimetric accuracy and precision ±0.2% (based on metal standards) and temperature accuracy ±0.5 °C. The initial mass used for the TG measurements was ~2.5 mg and the measurement was conducted in dry nitrogen.

TG-MS measurement. The same thermal analyzer coupled online with a Hiden Analytical HPR-20/QIC mass spectrometer (Warrington, United Kingdom) was used to perform the TG–MS measurements. The sample of newly synthesized zwitterionic compound C_1_C_4_imSO_3_ (~2.5 mg) was placed in an open alumina pan, the measurements were carried out in an argon atmosphere (flow rate: 100 cm^3^∙min^−1^), from room temperature to 450 °C, with a heating rate of 10 °C·min^−1^. The fragments were monitored between *m*/*z* = 1–100 through 30 channels in the Multiple Ion Detection (MID) mode with the electron impact ionization mode using an electron energy power of 70 eV. The collected MS data, RC RGA Analyzer, and MAS soft Manual Set were used for the TG–MS data collection.

DSC measurement. The differential scanning calorimetry (DSC) measurements for the zwitterionic compound C_1_C_4_imSO_3_ were recorded using a TA Instruments Differential Scanning Calorimeter DSC Q20 (TA Instruments, Milford, Massachusetts, USA) under an atmosphere of nitrogen (flow rate 50 cm^3^·min^−1^). The initial mass used for the DSC measurement was 5 mg. Specific heat capacity (*C*_p_) calibration was performed with sapphire crystal provided by TA Instruments as the reference material (sapphire heat material for hermetic pans, cylindrical shaped clear disk, 22 mg, 3.2 mm diameter and 0.4 thick). The standard temperature uncertainty was *u*(*T*) = ± 0.5 °C. Thermograms were recorded during cooling from from room temperature to −90 °C and during the reheating cycle to 300 °C, at a cooling and heating rate of 10 °C·min^−1^.

#### 3.2.3. Electrode Material Characterization (XRD and SEM)

XRD. The crystal structure of the electrode TiO_2_ nanotubes was examined by X-ray diffraction (XRD) collected with a Philips PW diffractometer 1050 with Cu-K*α*1,2 radiation in a 2θ range between 20 and 80° with step size 0.05° and counting time 2 s per step. Appendix A shows the XRD Ti foil patterns before and after anodization, where there are anatase (marked with asterisk) and Ti-metal phase reflections.

SEM. The morphology of the obtained TiO_2_ NTs surface was investigated using a JEOL JSM 6460LV scanning electron microscope (SEM). From the SEM images presented in Appendix A, we can see that the NTs have a more or less cylindrical shape with an outer diameter of ~150 nm, inner diameter of ~80 nm, and wall thickness of 35–40 nm.

#### 3.2.4. Flammability Test

The flammability of the LiTFSI/C_2_C_2_imTFSI electrolyte and LiTFSI/C_2_C_2_imTFSI/C_1_C_4_imSO_3_ electrolyte could be examined by directly observing the flame on the surface of the solution for 60 s, using a Digital Thermocouple Thermometer Dual-channel LCD Backlight Temperature Meter with an R-type Thermocouple sensor probe. The details about the mass measurement and burner exposition around 1200 °C are described in Papović et al. [11].

#### 3.2.5. Electrochemical Experiments

Galvanostatic measurement. Galvanostatic (GS) cycling was performed in a two-electrode bottle-type cell made of Pyrex glass, with a Teflon stopper closed with a double “O” ring. The transparent Pyrex glass cell filled with ~3 cm^3^ of electrolyte enables the monitoring of a possible change in the color of the electrolyte during cycling. The two-electrode cell was compiled in a glove box filled with argon. Li-metal foil was used as the counter electrode and the Ti/TiO_2_ NTA foil as the working electrode. The studied electrolyte was in contact with 1 cm^2^ of surface area of the working electrode. GS cycling was carried out at 25 °C by using the battery testing device Arbin BT 2042. GS cycling of the Ti/TiO_2_ NTA electrode was conducted using a current density of 50 µA·cm^−2^. The specific capacity was calculated by using the mass of the active electrode material, i.e., TiO_2_ NTs, which was obtained by scraping NTs from the Ti foil after all experiments were done.

## 4. Conclusions

The zwitterionic (ZI) compound C_1_C_4_imSO_3_ was synthesized to investigate its potential stabilizing effect on electrodes in LIBs during galvanostatic cycling, which corresponds to actual battery charging/discharging conditions. The specific thermophysical properties of pure C_1_C_4_imSO_3_ were examined by simultaneous TG, TG–MS, and DSC analysis, and showed that the compound has quite high stability. The electrochemical properties of a 0.5 M solution of LiTFSI in the LiTFSI/C_2_C_2_imTFSI/C_1_C_4_imSO_3_ system as a potential electrolyte for LIBs were tested and compared with the electrolyte LiTFSI/C_2_C_2_imTFSI, using a robust anatase TiO_2_ NTA electrode as the anode material. Despite the low vapor pressure, non-flammability and room temperature electrochemical stability of the ionic liquid-based electrolytes, indicated their possible use in safer LIBs and the need for their testing with functionalized additives was emphasized. In the case of the LiTFSI/C_2_C_2_imTFSI/C_1_C_4_imSO_3_ electrolyte, electrochemical galvanostatic experiments showed a significant improvement in capacity retention and Coulombic efficiency of the Ti/TiO_2_ NTA electrode, compared to the electrolyte without the ZI additive. According to IR spectroscopy measurements, the same surface sites of the nanotubes are responsible for the decomposition of the ZI additive and ionic liquid-based electrolyte during galvanostatic cycling. The decomposition of the ZI additive blocked those surface sites of the TiO_2_ nanotubes and thus prevented the decomposition of the LiTFSI/C_2_C_2_imTFSI ionic liquid-based electrolyte, which led to the improvement of the electrochemical properties.

## Figures and Tables

**Figure 1 ijms-24-03495-f001:**
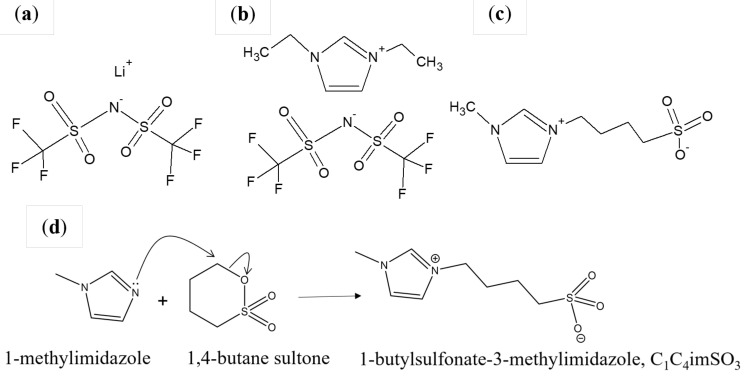
Chemical structures of: (**a**) lithium salt LiTFSI, (**b**) ionic liquid C_2_C_2_imTFSI, (**c**) zwitterionic additive C_1_C_4_imSO_3_, and (**d**) synthesis of zwitterionic additive C_1_C_4_imSO_3_.

**Figure 2 ijms-24-03495-f002:**
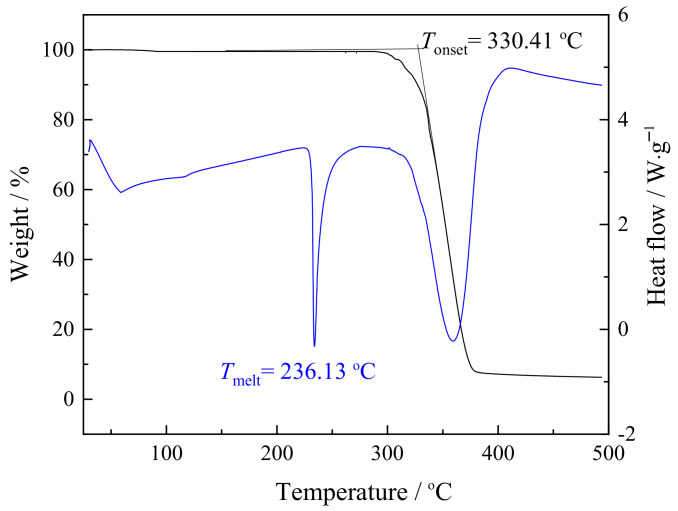
Simultaneous TG and DSC curves for synthesized C_1_C_4_imSO_3_.

**Figure 3 ijms-24-03495-f003:**
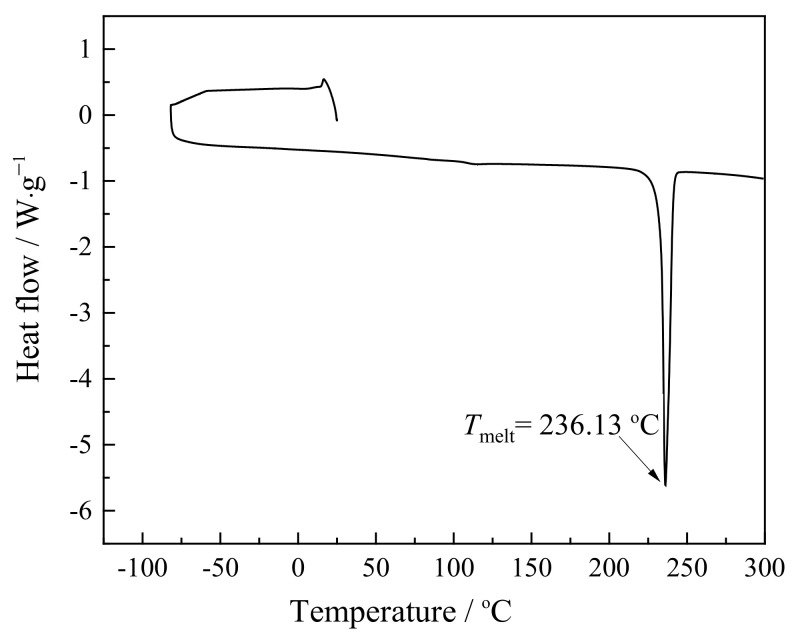
DSC curve for C_1_C_4_imSO_3_.

**Figure 4 ijms-24-03495-f004:**
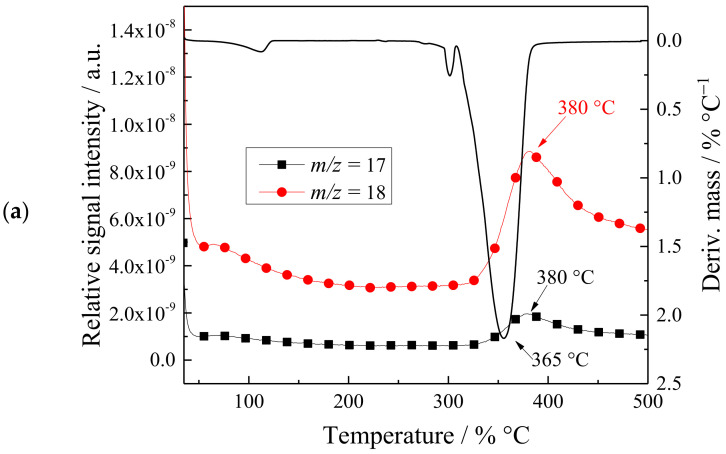
The DTG curves (black line) and TG–MS fragment ion intensities for C_1_C_4_imSO_3_ fragments *m*/*z =*: (**a**) 17 and 18, (**b**) 28 and 32, (**c**) 48 and 64, (**d**) 14 and 16, and (**e**) 36 and 44.

**Figure 5 ijms-24-03495-f005:**
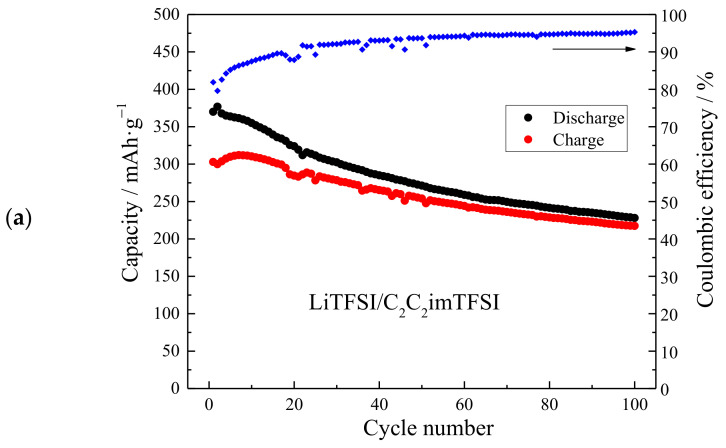
Galvanostatic discharge/charge performance with Coulombic efficiency (blue dot) of anatase TiO_2_ NTAs in: (**a**) LiTFSI/C_2_C_2_imTFSI electrolyte, (**b**) LiTFSI/C_2_C_2_imTFSI/C_1_C_4_imSO_3_ electrolyte, (**c**) direct comparison of capacities in both electrolytes, and (**d**) voltage profiles for 100th cycle for both electrolytes.

**Figure 6 ijms-24-03495-f006:**
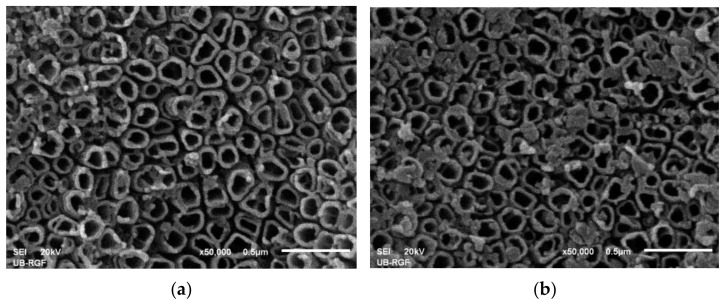
SEM images of TiO_2_ NTAs: (**a**) before cycling and (**b**) after cycling in the LiTFSI/C_2_C_2_imTFSI/C_1_C_4_imSO_3_ electrolyte.

**Figure 7 ijms-24-03495-f007:**
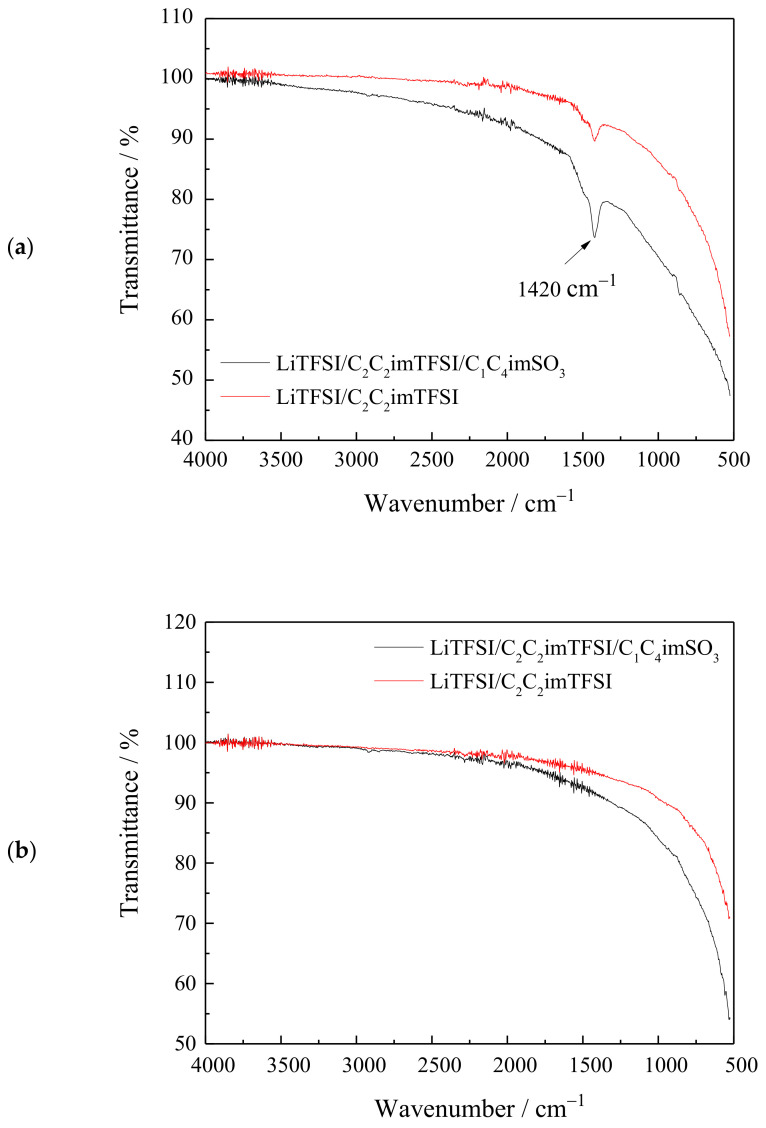
FTIR spectrum of Ti/TiO_2_ NTA electrodes in electrolytes LiTFSI/C_2_C_2_imTFSI/C_1_C_4_imSO_3_ or LiTFSI/C_2_C_2_imTFSI after GS cycling: (**a**) after washing with acetone and (**b**) after additional wash with water.

**Figure 8 ijms-24-03495-f008:**
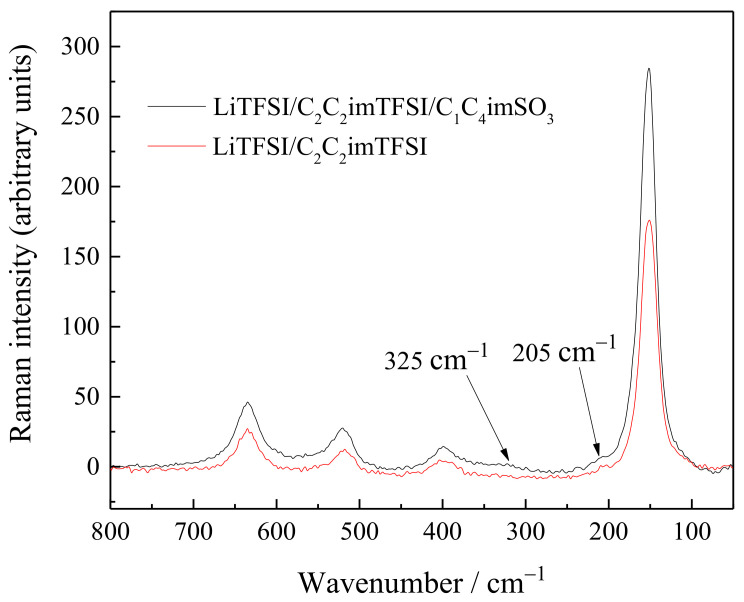
Raman spectra of electrode material after cycling in different electrolytes: LiTFSI/C_2_C_2_imTFSI and LiTFSI/C_2_C_2_imTFSI/C_1_C_4_imSO_3_.

**Table 1 ijms-24-03495-t001:** Purity and provenance of the chemicals used in experimental work.

Chemical Name	Provenance	Product Number	Purification Method	Final Mass Fraction	Water Content (ppm) ^b^
C_2_C_2_imTFSI ^a^	IoLiTech	174899-88-8	Vacuum drying	*ω* ≥ 0.99	13
LiTFSI ^c^	Sigma Aldrich	90076-65-6	Vacuum drying	*ω* ≥ 0.9995	-
1-methylimidazole	Sigma Aldrich	616-47-7	-	*ω* ≥ 0.99	18
1,4-butane sultone	Sigma Aldrich	1633-83-6	-	*ω* ≥ 0.99	20
C_1_C_4_imSO_3_	Synthesized in our laboratory		SLE ^e^, Vacuum drying	*ω* ≥ 0.99 ^d^	
Ti foil	Alfa Aesar	7440-32-6	-	*ω* ≥ 0.995	-
NH_4_F	Sigma Aldrich	12125-01-8	-	*ω* ≥ 0.995	-
Glycerol	Sigma Aldrich	56-81-5	-	*ω* ≥ 0.995	-
Indium calibration standard	Thermal Analysis Instruments	-	-	*ω* ≥ 0.9999	-
Sapphire specific heat material for hermetic pans	Thermal Analysis Instruments	-	-	*ω* ≥ 0.9999	-
Acetonitrile	Sigma Aldrich	75-05-8	-	*ω* ≥ 0.998	-

^a^ C_2_C_2_imTFSI = 1,3-diethylimidazolium *bis*(trifluoromethylsulfonyl)imide. ^b^ KF titration = Karl Fischer titration. ^c^ LiTFSI = lithium *bis*(trifluoromethylsulfonyl)imide. ^d^ determined by NMR. ^e^ SLE = solid–liquid extraction.

## Data Availability

Not applicable.

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
