# Peer review of "Effect of Zwitterionic Additive on Electrode Protection through Electrochemical Performances of Anatase TiO2 Nanotube Array Electrode in Ionic Liquid Electrolyte"

_ijms, 2023, doi:10.3390/ijms24043495_

Round 1

Reviewer 1 Report

The manuscript reports on the characterization of Tio2 nanotubes in a zwitterionic compound for potential application in lithium ion batteries.
Interesting results are presented and the compound may be promising.
However, a discussion and comparison of the results with the literature was not presented
to enable further development of the zwitterionic compound.
Please note there is a confusion regarding the order in the presentation of results and figures,
that should be rectified.
Please also ensure the order of the results and experimental agree with the requirements of the journal.
The size of the fonts is not uniform throughout the manuscript as indicated in the attached pdf file.

Author Response

Revision

Journal IJMS (ISSN 1422-0067)

Manuscript ID ijms-2183421

Type Article

Effect of additive on electrode protection through electrochemical performances of anatase TiO2 nanotube arrays electrode in ionic liquid electrolyte

Reviewer 1

The manuscript reports on the characterization of Tio2 nanotubes in a zwitterionic compound for potential application in lithium ion batteries.
Interesting results are presented and the compound may be promising.
However, a discussion and comparison of the results with the literature was not presented
to enable further development of the zwitterionic compound.
Please note there is a confusion regarding the order in the presentation of results and figures,
that should be rectified.
Please also ensure the order of the results and experimental agree with the requirements of the journal.
The size of the fonts is not uniform throughout the manuscript as indicated in the attached pdf file.

Thank you very much for all your sugestions and questions. We hope your questions are answered as far as we can, suggestions are accepted, and the Manuscript has been improved according to them.

Page 2, rows 45 and 46. The rationale for selecting TiO2 nanotube arrays as electrodes is not clear. A paragraph introducing and comparing the literature on efficient and suitable nanotube electrodes can address this issue.

Thank you for suggestion. We tried to make more clear the using of TiO2 nanotube arrays in this work, described in Introduction part of the Manuscript, with adequate references.

TiO2 nanotubes are specific material. That kind of electrode combines current collector and electrode material. TiO2 is robust and resistant material which allows working on higher temperatures because in that material there is no possibility of removing material from the electrodes (adhesion) at higher temperatures. Also, such NTA electrodes enable immediate, direct recording of Raman and FTIR spectra on their surfaceIn NTAs Li transport properties are geometrically enhanced. Most of Li-based organic electrolytes work well with this material due to shortening of the diffusion length from the nanostructure.

Page 2, rows 50 and 59. What are the simultaneous duties? Expanding to these more specifically is required to adequately introduce the novelty of the work.

Moreover, a figure introducing the studied compounds, structures and processes relevant to batteries is missing to aid the reader grasping the presented research.

Introduction of figures of the investigated compounds for chemical processes is done at Figure 1, where are presented all investigated compounds, as well shown the synthetic path of new compound which served for functionalization of used electrode during the experiments.

Page 2, row 77. Do the authors meant figure 1?

The graph data cannot be easily discerned. Using a thicker line or higher quality figure is suggested. The same applies to the next figure (3).

The Authors meant Figure 2, because we put the investigated compounds' structures in Figure 1. Because the graph data could not be easily discerned, we made the graph from Figure 2 and Figure 3 in the Origin file. The higher quality of Figure 2 and Figure 3 is achieved by replacing the previous Figure with one made in the Origin programme, with thicker lines.

Page 2, rows 80 and 81. A few % reduction from 100oC to 125oC can be seen. No disucssion of this was added in the manuscript. Can this be relevant to water?

Because a few % of reduction from 100 oC to 125 oC is seen in previous version of the Manuscript, we dried the sample at a temperature 140 oC, and then we measured thermal stability which is now presented in Figure 2.

Page 3, rows 86 to 88. What is the spacer length? Explicitly quantifing will allow for a comparison with other research.

Spacer length is the hydrocarbon chain that is connecting, in case of our studied zwitterion, the imidazole ring and the sulfonate group and the spacer is hydrocarbon part which is connecting them. With the comparison with compounds based on sulfonate anion and imidazole cation part, presented in the work Yoshizava et al. [31] and Galin et al. [36], where is only difference in length of the molecule part that connects them, could be noted that longer spacer indicate lower melting points. This is probably because of the flexibility increase with spacer increasment.

Page 3, rows 109 and 110. The letters are now consistent with others.

Page 5, rows 122 and 123. It is difficult to follow the TG-MS results and their interpretation scattered in four different plots. Unless there is a specific reason for their separate presentation, these will be better presented in one plot of ascending m/z order. Alternatively, each plot can be grouped according to the discussion of each component.

We tried to present on the TG-MS graphs precisely because of the different responses those that are of a similar order of size and that are separated in the form of fragments in similar temperature intervals and mechanical steps. That is why it is presented in several small graphs. And in the table in Supporting material, the exact size order of the fragments is given (exactly Table S1; Summarized the identification results and normalized ion currents for an investigated sample of synthesized zwitterionic compound C1C4imSO3.).

Page 6, rows 131 and 132. The discussion of previous TGC-DSC results suggested a not hygroscopic nature, which contradicts this comment. Please rephrase or explain.

The zwitterionic compound is not hygroscopic, and the TG-MS results show regroup of atoms into water molecules at elevated temperatures, such as 350 oC and more, as shown in the Figure 4a.

Page 6, rows 133 and 134. The letters are now consistent with others.

Page 6, rows 160 and 161. A reader woudl normally expect S1-S4 before this figure. The flammability experiment could also be moved after the structural characterization.

Actually, to clarify why the error occurred. Firstly the Manuscript was sent where it was as part “2. Experimental work, materials and apparatus” and according to that, we have Figures S1-S4 in that part. Then was comming part regarding the Results and Discussion, where was Figure S5. But, the editorial of the Journal replaced Experimental work, materials and apparatus to the end of Manuscript, and all Figures were shifted. Now, the revised form of the Manuscript is prepared well, now is Figure S1.

Page 7, rows 193 and 194. The colour results were not shown. These can either be added inset in figure 6 or clarified in text. 

That change of colour is previously presented in Figure S6. Actually, following this your comment and your later comment, it is added the content of Figure S6 to the revised version of the Manuscript in Figure 6c, with entitling the Figure as “c) comparison of capacity between them with appearance of electrolyte at the end of cycling”.

Page 7, rows 202 and 203. This cannot be seen in the separate figures. Plotting in the same figure can address this issue.

Thank you for your comment. In this revised version of the Manuscript it is plotted in the same Figure as the part of the Figure 6c, for better comparison to address this issue. We left the separate figures as Figure 6a and Figure 6b because here we are mentioning the Coulomb efficiency for each system in the text and to be shown in the Figures. Please, see the attachment. Here is Figure presented, in this Answer.

b)

a)

d)

c)

d)

Figure 6. Galvanostatic discharge/charge performance with Coulombic efficiency of anatase TiO2 NTAs in: a) LiTFSI/C2C2imTFSI electrolyte, b) LiTFSI/C2C2imTFSI/C1C4imSO3 electrolyte, c) comparison of capacity between them with appearance of electrolyte at the end of cycling, as well as d) voltage profiles for 100th cycle for both electrolytes.

Page 7, row 206. The incorrect word “ramains” is replaced with the grammatically corrected word “remains”.

Page 9, row 234. Please plot in the same figure to enable a comparison.

Thank you for your comment. In this revised version of the Manuscript it is plotted in the same Figure as the part of the Figure 6c, for better comparison to address this issue.

Page 9, rows 238 and 242. Adding figure S6 inset in figure 6b is suggested to improve the presentation of the results.

Thank you for your comment. Now is added the content of previous Figure S6 to the revised version of the Manuscript in Figure 6c, and mentioned in the text of the Manuscript.

Page 10, row 268. Figure S7 shows uv/vis and not vibrational spectra.

Thank you for your careful comment. Of course, the vibrational spectroscopy from this part of the text is deleted.

Page 10, row 283. The letters are now consistent.

Page 10, rows 283 and 286. Please indicate the peaks in the figure. Plotting a and b in the same plot is also required for comparison of the FTIR spectra.

Thank you for your comment. In this revised version of the Manuscript the peaks are plotted in the same Figure, for better comparison of FTIR spectra.

Page 11, rows 287 and 292. All spectra of figure S8 can be overlapped in figure 8 for a better presentation of the discussed evidence. A uniform font size should also be used.

Thank you for your notification. A uniform font size is used. The spectra from previous Figure S8 are overlapped in Figure 7, as part of Figure 7 (exactly Figure 7b).

Figure 7. FTIR was recorded in the range from 500–4000 cm–1 at room temperature. Spectra for electrodes surface: a) after washing with acetone, which was in electrolytes LiTFSI/C2C2imTFSI/C1C4imSO3 or LiTFSI/C2C2imTFSI and b) after acetone, additionaly washed with water; FTIR spectra were recorded after galvanostatic cycling.

Page 12, rows 300 and 304. Please indicate the vibration types in the plots. Similar to previous comments all spectra can be plot in the same graph and properly compared.

The font size is again not uniform.

Thank you for your notification. The uniform font size is used now. The spectra from Figure 9a and Figure 9b overlapped in Figure 8 (shifted number) and the vibration types are indicaed in the plots.

Page 13, rows 313 and 316. A comparison with reported structures in the literature is missing based on the presented results. Some of these reports have been presented in the introduction, but not adequately compared here. This discussion can also suggest future steps for further development of efficient ionic compounds. Aditional literature on ionic surfactants with nanotubes to be compared includes: https://doi.org/10.1021/acs.jpcc.6b06272

https://doi.org/10.1021/acs.jpcc.9b03341

Thank you for your insightful comment. Based on the presented results, the comparison with reported structures from the literature is made in this part of Manuscript. Also, the additional literature on ionic surfactants with nanotubes is added and compared with the presented results.

To suggest the future steps for further development of efficient zwitterionic compound, the discussion is extended regarding to the compounds with the similar structure, such as SDS (which contains also the SO3 group).

  1. Lutsyk, Y. Piryatinski, M. Shandura, M. A. Araimi, M. Tesa, G. E. Arnaoutakis, A. Ashwin Melvin, Ol. Kachkovsky, A. Verbitsky, A. Rozhin, Self-Assembly for Two Types of J-Aggregates: cis-Isomers of Dye on the Carbon Nanotube Surface and Free Aggregates of Dye trans-Isomers, J. Phys. Chem. C 2019, 123, 32, 19903-19911.

DOI: 10.1021/acs.jpcc.9b03341

  1. Lutsyk, Y. Piryatinski, M. A. Araimi, R. Arif, M. Shandura, O. Kachkovsky, A. Verbitsky, A. Rozhin, Emergence of Additional Visible-Range Photoluminescence Due to Aggregation of Cyanine Dye: Astraphloxin on Carbon Nanotubes Dispersed with Anionic Surfactant, J. Phys. Chem. C 2016, 120, 36, 20378-20386.

DOI: 0.1021/acs.jpcc.6b06272

In some reported papers, the previously investigated zwitterionic compounds, such as N,N-dimethylpyrrolidinium methyl sulfonate show the minimization of electrolyte decomposition and resulting in interfacial stability of used electrodes [38]. Also, some of zwitterionic compounds are 1-(1-Butylpyrrolidinium)butane-4-sulfonate betaine, 1-(Tri-n-butylphosphonium)butane-4-sulfonate betaine [54], contributing to the stabilization of the Li+ ions in the mixtures. Compounds with similar sulfonate functional group, such as sodium dodecylbenzene sulfonate [55] based on their interaction with nanotubes show how the self-organization of organic molecules with carbon nanotubes leads to the formation of functionalized molecular structures with advanced properties [55,56]. Some future steps for further development of efficient zwitterionic compounds could lie in the modification of substituent of imidazole ring (such as vinyl group, or oxygenated alkyl side chain attached at one N in the imidazole ring) with the same SO3 part from the other side of imidazole ring. The interfaces’ structure and the electrode-protecting films' nature are other aspects that need appropriate consideration.

Page 13, rows 323 and 325. The application of the compound to the nanotubes was not described. Please explain.

The zwitterionic compound is added to the used electrolyte inside the cell where TiO2 nanotubes were electrode, and during the cycling experiment, the zwitterionic compound is partially adsorbed at the surface of the electrode made of nanotubes. So, the zwitterionic compound is not previously added at the surface of TiO2 nanotubes. They were prepared as described, and during and after the experiment, it was spontaneously remained at the surface.

Page 13, row 338. Very informative table. A reference to each compound in the list may also be added.

Thank you for comment. Actually, we have made the changes, because structures of investigated compounds we put all together in Figure 1. The part of table that was relied at structures of chemicals is lefted out, because we put the most important (investigated structures) in Figure 1.

Page 15, rows 393. Please define the acronyms in their first instance in the manuscript.

Thank you for your valuable input. Since the “MS” is first occurring here, the explanation of that abbreviation is added. Not only for the abbreviation “MS”, but we have also done that for all abbreviations that we have noticed throughout the whole Manuscript. For example, after the Abstract, first instance in the manuscript is the title 2.2, which is not more TG-MS, it is now “Thermogravimetry coupled with mass spectrometric (TG-MS) analysis”.

PLEASE, see the attachment.

Reviewer 2 Report

The authors synthesized zwitterionic compound C1C4imSO3 that may be of use for the lithium-ion batteries production. They tested C1C4imSO3 in the solution of LiTFSI in ionic liquid C2C2imTFSI using anatase TiO2 nanotube arrays electrode as anode material and found that the addition of 3% C1C4imSO3 improves lithium-ion intercalate/deintercalate properties. The structure and purity of synthesized C1C4imSO3 were confirmed by NMR and FTIR measurements, while the thermal stability of the pure C1C4imSO3 was examined by simultaneous TG and DSC measurements, as well as by TG-MS technique.  In general, this work represents a solid practical investigation and deserves publishing in IJMS.

However, the paper is hard to read and requires a modification of style.  For example, the first phrase in the abstract is not grammatically correct:

“In this work is a tested electrolyte based on synthesized functionalized zwitterionic (ZI) additive 1-butylsulfonate-3-methylimidazole C1C4imSO3 in 0.5 M solution of LiTFSI in ionic liquid (IL) 1,3-diethylimidazolium bis(trifluoromethylsulfonyl)imide, C2C2imTFSI”.

Maybe, it would be better to write “An electrolyte based on the synthesized functionalized zwitterionic … was(is) tested in this work.” Such stylistic inconsistences appear from time to time throughout the whole text. Please, pay attention to this issue.

The phrase “From the presented thermogravimetric curve (Figure 2, blue line), synthesized C1C4imSO3 is stable at a temperature of at least 330 °C, indicating its good thermal stability” in lines 78-81 should actually be addressed to a green line in Fig. 2 in the first place. Even more so, this phrase also looks incomplete. Maybe it should start as follows: “From the presented thermogravimetric curve (Figure 2, green line) it follows that ….”

            The structures of compounds are presented as far as in 13th page (experimental part). This is not convenient. Please provide the main structures somewhere in the beginning.   

Author Response

Revision

Journal IJMS (ISSN 1422-0067)

Manuscript ID ijms-2183421

Type Article

Effect of additive on electrode protection through electrochemical performances of anatase TiO2 nanotube arrays electrode in ionic liquid electrolyte

Reviewer 2

The authors synthesized zwitterionic compound C1C4imSO3 that may be of use for the lithium-ion batteries production. They tested C1C4imSO3 in the solution of LiTFSI in ionic liquid C2C2imTFSI using anatase TiO2 nanotube arrays electrode as anode material and found that the addition of 3% C1C4imSO3 improves lithium-ion intercalate/deintercalate properties. The structure and purity of synthesized C1C4imSO3 were confirmed by NMR and FTIR measurements, while the thermal stability of the pure C1C4imSO3 was examined by simultaneous TG and DSC measurements, as well as by TG-MS technique.  In general, this work represents a solid practical investigation and deserves publishing in IJMS.

However, the paper is hard to read and requires a modification of style.  For example, the first phrase in the abstract is not grammatically correct:

“In this work is a tested electrolyte based on synthesized functionalized zwitterionic (ZI) additive 1-butylsulfonate-3-methylimidazole C1C4imSO3 in 0.5 M solution of LiTFSI in ionic liquid (IL) 1,3-diethylimidazolium bis(trifluoromethylsulfonyl)imide, C2C2imTFSI”.

We agree with the Reviewer's suggestions contained in this point of the review. We have corrected grammatically made errors throughout in the revised form of the Manuscript.

Maybe, it would be better to write “An electrolyte based on the synthesized functionalized zwitterionic … was(is) tested in this work.” Such stylistic inconsistences appear from time to time throughout the whole text. Please, pay attention to this issue.

Thank you for your notification. We have now payed more attanetion to this stylisic inconsistences and hopefully that we have corrected them.

The phrase “From the presented thermogravimetric curve (Figure 2, blue line), synthesized C1C4imSO3 is stable at a temperature of at least 330 °C, indicating its good thermal stability” in lines 78-81 should actually be addressed to a green line in Fig. 2 in the first place. Even more so, this phrase also looks incomplete. Maybe it should start as follows: “From the presented thermogravimetric curve (Figure 2, green line) it follows that ….”

            The structures of compounds are presented as far as in 13th page (experimental part). This is not convenient. Please provide the main structures somewhere in the beginning.  

Thank you for your notification. The missing chemical structures of the studied compounds are added at the beginning, as Figure 1.

Thank to Reviewer 2 very much for all sugestions and questions. We hope that yours questions are answered as far as we can, suggestions are accepted and the Manuscript has been improved according to them.

PLEASE, see the attachment.

Round 2

Reviewer 1 Report

The authors applied significant revisions to the original manuscript. As a result the manuscript has been greatly improved and the presentation of the results is now easier to follow. Please make sure replies to responses are added in the manuscript, e.g. comment in page 13, rows 323-325

Author Response

Revision

Journal IJMS (ISSN 1422-0067)

Manuscript ID ijms-2183421

Type Article

Effect of additive on electrode protection through electrochemical performances of anatase TiO2 nanotube arrays electrode in ionic liquid electrolyte

Reviewer 1

The authors applied significant revisions to the original manuscript. As a result the manuscript has been greatly improved and the presentation of the results is now easier to follow. Please make sure replies to responses are added in the manuscript, e.g. comment in page 13, rows 323-325

Thank you very much for respecting ours improvements through the text of the Manuscript. Thank you for all your suggestions and questions and by them, the assistance provided to make presentation of the results now easier to follow. We hope your question (specifically Page 13, rows 323 and 325) is answered as far as we can, suggestion to add it to the main text is accepted and added, and the Manuscript has been improved according to them.

Page 13, rows 323 and 325. The application of the compound to the nanotubes was not described. Please explain.

The zwitterionic compound based on imidazole ring and sulfonate group is added to the investigated electrolyte inside the cell where TiO2 nanotubes were electrode, and during the cycling experiment, the zwitterionic compound is partially adsorbed at the surface of the electrode made of NTAs. So, the zwitterionic compound is not previously added at the surface of TiO2 nanotubes. TiO2 nanotubes were prepared as described in the text, and during and after the experiment, the zwitterionic compound was spontaneously remained at the surface.

It should be noted that after galvanostatic cycling experimens (described in section 2.4.2) the SO3 group from C1C4imSO3 zwitterionic compound from the electrolyte (detailed electrolyte content is described in next section 3.1.1) is partially adsorbed at the TiO2 nanotubes used as electrode in these experiments.

The explanation of adding the zwitterionic compound to the electrolyte is described by this sentence in the part “3. Experimental part / 3.1. Materials / 3.1.1. Synthetic procedure for the zwitterionic compound” other investigated electrolyte was obtained by dissolving 0.5 moldm-3 LiTFSI in a previously made mixture of ionic liquid C2C2imTFSI and ZI compound C1C4imSO3, where is added 3% of C1C4imSO3.
